# Healthcare professionals interpersonal variability and determinants of medical decision thresholds for active management of extremely preterm infants in a level 3 perinatal center in France

**Charlotte Girard** [1]*, **Hélène Collinot**[1,2], **Héloïse Torchin**[3,4], **Clara Rollet**[1,5], **Pierre-Henri Jarreau**[3,4], **François Goffinet**[1,3]

**1** Maternité Port Royal, Cochin-Broca-Hôtel Dieu Hospitals, Université Paris Cité, Assistance Publique-Hôpitaux de Paris, DHU Risk in Pregnancy, Paris, France, **2** Equipe "From Gamete To Birth", Institut Cochin, Université Paris Cité, CNRS UMR, INSERM U1016, Paris, France, **3** EPOPé, CRESS U1153, Université Paris Cité, Université Sorbonne Paris Nord, Inserm, INRAE, Paris, France, **4** Neonatal Intensive Care Unit, Cochin-Broca-Hôtel Dieu Hospitals, Université Paris Cité, Assistance Publique-Hôpitaux de Paris, Paris, France, **5** Midwifery School of Baudelocque, Université Paris Cité, Assistance Publique-Hôpitaux de Paris, Paris, France

* charlotte.girard4@gmail.com

## Abstract

### Background

Before 26 weeks of gestational age, because extremely preterm infants (EPI) face a high risk of death or disability, management decisions may involve either active treatment or palliative care. Survival chances largely depend on the willingness of medical teams and parents to opt for active management. Variability of practices explains differences in survival between countries and regions, and interpersonal variability may also exist among caregivers within the same center. Our objective was to study the variability of management decisions and their determinants among caregivers in a French type 3 maternity hospital.

### Methods

All caregivers, obstetricians, pediatricians, and midwives, involved in the management of EPI in a type 3 perinatal center were surveyed using a self-administered questionnaire. Each respondent reported their personal thresholds for deciding on active management, defined as the unborn child's estimated likelihood of survival without severe neonatal morbidity. Median and interquartile ranges (IQR) of these thresholds were calculated and compared by respondent characteristics.

### Results

85 (75%) eligible professionals responded. The median threshold of survival without severe neonatal morbidity below which active management was deemed impossible was 15% (IQR 10-30%), while the median threshold above which active management could

**Data availability statement:** All raw data and code files are available from the Zenodo database (DOI:10.5281/zenodo.11187600).

**Funding:** The author(s) received no specific funding for this work.

**Competing interests:** The authors have declared that no competing interests exist.

not be refused was 80% (IQR 70-90%). Wide IQRs indicated significant variability in individual thresholds. This variability appeared to be influenced by profession and gender but was not associated with factors such as having children, age, experience, or the personal estimates of the neonates' outcomes.

## Conclusions

Decision thresholds for active management of EPI, expressed in terms of survival without severe neonatal morbidity, vary significantly among professionals. The thresholds reported in our study were notably higher than those observed in other countries, which may help explain the lower rates of active management before 26 weeks in France. Recognizing these differences and comparing personal thresholds with peers could facilitate more consensus-based decision-making within teams.

## Introduction

Extremely preterm infants (EPI), born between 22 and 28 weeks of gestation (WG), account for around 0.4% of births [1]. They can survive only if they receive active obstetric-pediatric management, and remain at high risk of perinatal death, severe complications, and long-term disabilities [2]. Wealthy countries concur in offering routine active management starting at 26 weeks for these children whose outcome is considered favorable [3,4]. For children born between $22^{0-7}$ and $25^{6-7}$ weeks, obstetric-pediatric teams and parents face a difficult ethical choice [5–7]. They can either institute active management, accepting invasive care and a risk of the child's death or disability, or decide on palliative care that ends in death [8], accepting the risk of an opportunity loss. These children's survival thus depends first on the willingness to begin active management [7,9,10]. In France, active management for children born before 26 weeks is offered less often than in other countries of similar socioeconomic level: it occurs very frequently starting at 26 weeks, is considered at 23-25 weeks, and not offered before 23 weeks [7,11]. In Sweden, Great Britain, and the USA, it is very frequent by 24 weeks and considered at 22 and 23 weeks [3,12,13]. This difference probably explains the lower survival rate of children born before 26 weeks in France than elsewhere [14–16].

No scientific data enable clinicians to distinguish without a doubt the situations requiring active management from those requiring palliative care. At the moment of decision, each professional and each team have their own criteria: objective clinical data, but probably also more subjective factors, such as team practices, professional's knowledge of management and outcome, sex, experience, perception of disabilities, and religion among others [17–19].

Recommendations issued by professional societies can help guide this decision-making. These differ between countries; some offer standardized — and therefore equal — care nationwide, while others advocate an individualized decision that therefore depends more on the professional at hands [11–13,20]. This probably explains the great heterogeneity of the decision to undertake active management in France, with its rates ranging from 20% to 80% by region [21]. This geographic variability raises ethical questions about equitable access to care. Individual variability in these decisions has been little studied — nationally, regionally, or even within individual teams, and these few publications have not studied the determinants explaining these differences [17–19,22].

In France and elsewhere, the clinical criterion most often used to guide this decision is a gestational age threshold, but many other factors, including birthweight, sex, and comorbidities are also linked to the outcome of EPI [23]. Logically, therefore, the decision should be

based on an assessment of the child's future health rather than exclusively on gestational age at birth. Wilkinson et al. have proposed basing decision-making on a prognostic threshold expressed as the theoretical likelihood of survival without severe disability. Their study was the first in which neonatal physicians' views about prognosis-based thresholds for resuscitation were assessed and they did not issue any recommendations [24]. Such thresholds are not routinely used and have never been assessed in France.

Our study's objective was to describe the variability in professionals' individual thresholds for choosing active management and to identify their determinants. To achieve this, we adopted an approach similar to that of Wilkinson et al., basing decision-making on prognostic thresholds expressed as survival without severe disability. We applied this framework to the management decisions for EPI (born before 26 WG) in a single obstetrics-pediatric unit.

## Methods

This prospective, observational single-center survey was conducted among professionals providing obstetric and neonatal care at a French level 3 perinatal center. Data were collected by a self-administered questionnaire (S9 Appendix) completed in the workplace, with an investigator present. The questionnaires were completed and collected over a 2-month period from February 1 through April 1, 2020.

This study's target population comprised the professional staff at the Port-Royal Maternity Hospital (Paris) regularly involved in managing situations at risk of extremely preterm birth (EPB): obstetricians and pediatricians (residents and hospitalists for both groups), and staff midwives. We included only staff who worked more than half time at Port-Royal and practiced actively during the study period. We collected participants' sex (male or female), whether or not they had any children, their professional category (midwife, obstetrician, or pediatrician), hospital status (staff midwife, staff physician, resident), age, and professional experience (number of years since completion of their degree or, for residents, since their residency began). We analyzed the latter two characteristics in three categories: age as less than 30 years, 30–40 years, and older than 40 years, and professional as less than 2 years, 2–10 years, and longer than 10 years. The categories were defined to ensure homogeneous group sizes, with the aim of increasing statistical power.

In our maternity ward, management of situations at risk of EPB follows the EXPRIM protocol (EXtrem PRematurity Innovative Management) [25], established in 2015 and inspired by the Swiss guidelines [26]. For each case, the obstetric-pediatric team assesses the likely outcome of the child to be born during a group discussion in a non-emergency setting and chooses active or palliative management should the child be born in the next few days. This choice is based on several criteria including gestational age, estimated birthweight, fetal vital status, and any in utero growth restriction or chorioamnionitis. The protocol distinguishes four different attitudes toward active management that allow us to combine the proposal by the obstetric-pediatric team and the parents' desires (Fig 1): active management impossible (deemed futile for our team), unreasonable, reasonable, or systematic. At the meeting's end, an obstetrician and a pediatrician discuss with the parents the child's estimated risk and the proposed management. Final decision for active or palliative management is consensual, shared between medical team and parents.

### Principal objective – Individual decision thresholds for active management

We chose to examine management choices by using decision thresholds based on the fetus's likely outcome. Each participant had to define the theoretical likelihood of survival without severe morbidity (as a percentage), corresponding to their personal threshold for deciding on active management in EPB. The questionnaire applied to a theoretical population of children with a risk of birth between $23^{0/7}$ weeks and $25^{6/7}$ weeks, with no known major malformation,

| Attitudes toward active management | Impossible | Unreasonable | Reasonable | Systematic |
|---|---|---|---|---|
| Medical team proposal | Systematic palliative care | Active management inadvisable but acceptable if parents want it | Active management advised but parental request for palliative care acceptable | Systematic active management |
| Parental participation | Parents cannot demand care deemed excessive | The parents' request should be respected | The parents' request should be respected | Parents cannot refuse this care, in the best interests of the child |

Fig 1. Attitudes toward active management of an extremely preterm infant, defined by the EXPRIM protocol.

who had received at least one dose of corticosteroids before birth and with an estimated fetal weight greater than 500 g. Participants were asked to set four decision thresholds, according to the four attitudes from the EXPRIM protocol (Fig 1):

- 2 thresholds below which respondents would not choose active management (and would therefore choose palliative care), one if active management appeared impossible, and one if it appeared unreasonable;

- 2 thresholds above which respondents would choose active management, one if active management appeared reasonable and one if it should be systematic.

Respondents' decision thresholds were expressed as a theoretical likelihood of survival without severe morbidity.

Severe neonatal morbidity was defined, as in the Epipage-2 study [2], by at least one of the following conditions: cystic periventricular leukomalacia, intraventricular hemorrhage grades III and IV, retinopathy of the preterm grade III and/or requiring laser management, severe necrotizing enterocolitis and moderate or severe bronchopulmonary dysplasia.

## Secondary objectives — Associations between individual determinants and decision thresholds

We then studied decision thresholds as a function of respondents' individual determinants such as age, sex, etc.

We also collected — as an individual determinant — professionals' estimates of the infants' short- and medium- term outcomes. Specifically, the second part of the questionnaire asked participants for their estimates of the likelihood of survival without severe morbidity (expressed as a percentage) of an EPI managed actively in this unit during this period, for each of the following five clinical situations:

- Birth at $23^{0/7}$–$23^{3/7}$ weeks and birthweight 500–600 g.

- Birth at $23^{4/7}$–$23^{6/7}$ weeks, and birthweight 500–600 g.

- Birth at $24^{0/7}$–$24^{3/7}$ weeks and birthweight 600–700 g.

- Birth at $24^{4/7}$–$24^{6/7}$ weeks and birthweight 600–700 g.

- Birth at $25^{0/6}$–$25^{6/7}$ weeks and birthweight 700 g.

## Statistical analyses

Categorical variables (sex, at least one child, professional category, hospital function, age (<30 years, 30-40 years, >40 years) and experience (<2 years, 2-10 years, >10 years)) are presented as percentages.

Outcome estimates for EPI — collected as an individual determinant — are reported as medians with their interquartile ranges (IQRs) for each of the five clinical situations considered.

Decision thresholds are presented as medians with their IQRs, for each of the four attitudes. These were first calculated for the overall population study and then according to the different individual determinants.

Decision thresholds were then compared for each individual determinant for each attitude, with the appropriate nonparametric tests (Mann-Whitney for binary variables and Kruskal-Wallis for the categorical variables with more than two categories).

Given the bias due to the almost exclusively female midwife population, the study by sex was performed both including and excluding midwives.

To study the association between outcome estimates and management decisions, we compared the staff's decision thresholds by their outcome estimates. For each of the five clinical situations, we divided the respondents into four groups corresponding to the four quartiles of their outcome estimates. For each clinical situation the first group comprises the most pessimistic respondents and the fourth the most optimistic. For each group and each clinical situation, we reported decision thresholds as medians with their IQRs for each of the four attitudes. Finally, decision thresholds according to outcome estimate were compared with a Kruskal-Wallis test.

## Ethics approval and consent to participate

With regard to ethics approval, the " Comité d'Ethique de la Recherche - CER U-Paris Cité " studied our protocol and certified that this type of study is not required by French law to be submitted to an ethics committee.

With regard to privacy impact assessment, this study has been approved and registered by the " Registre général des traitements de l'APHP " under number: 2024 0412174206.

Consent was obtained orally from each participant by the principal investigator prior to answering the questionnaire. An information and non-opposition notice for the use of data was written with the project and was made available to the participants. This notice was provided to the research ethics committee that studied our protocol and contributed to its evaluation.

## Results

Among 115 professionals eligible, 86 (75%) participated. Questionnaires were completed by 23 (100% of those eligible) obstetrician-gynecologists, 18 (67%) pediatricians, and 45 (69%) midwives. Aberrant responses led to the exclusion of one respondent (midwife).

Among the 85 respondents, 73 (85.9%) were women. The median age was 33 years (IQR 28-42 years). In all, 44 (51.8%) were midwives, 23 (27.0%) obstetricians, and 18 (21.2%) pediatricians (Table 1).

## Individual decision thresholds for active management

The median threshold of survival without severe morbidity below which active management was considered impossible was 15% (IQR 10%-30%) (Fig 2). One quarter of the respondents

**Table 1. Respondents' individual characteristics.**

| Characteristics – N = 85 | n/N (%) |
|---|---|
| **Sex (woman)** | 73/85 (85.9) |
| **Age (in categories)** | |
| <30 years | 33/85 (38.8) |
| 30-40 years | 29/85 (34.1) |
| >30-40 years | 23/85 (27.1) |
| **Does not have a child** | 58/85 (68.2) |
| **Professional category** | |
| Obstetrician | 23/85 (27.0) |
| Pediatrician | 18/85 (21.2) |
| Midwife | 44/85 (51.8) |
| **Status** | |
| Resident | 14/85 (16.4) |
| Staff physician | 27/85 (31.8) |
| Staff midwife | 44/85 (51.8) |
| **Professional experience (in categories)** | |
| < 2 years | 18/85 (21.2) |
| 2–10 years | 39/85 (45.9) |
| > 10 years | 28/85 (32.9) |

1. Below what percentage likelihood of the infant's survival without severe morbidity would you consider that active management is impossible?
2. Below what percentage likelihood of the infant's survival without severe morbidity would you consider that active management is unreasonable?
3. Above what percentage likelihood of the infant's survival without severe morbidity would you consider that active management is reasonable?
4. Above what percentage likelihood of the infant's survival without severe morbidity would you consider that active management should be systematic?

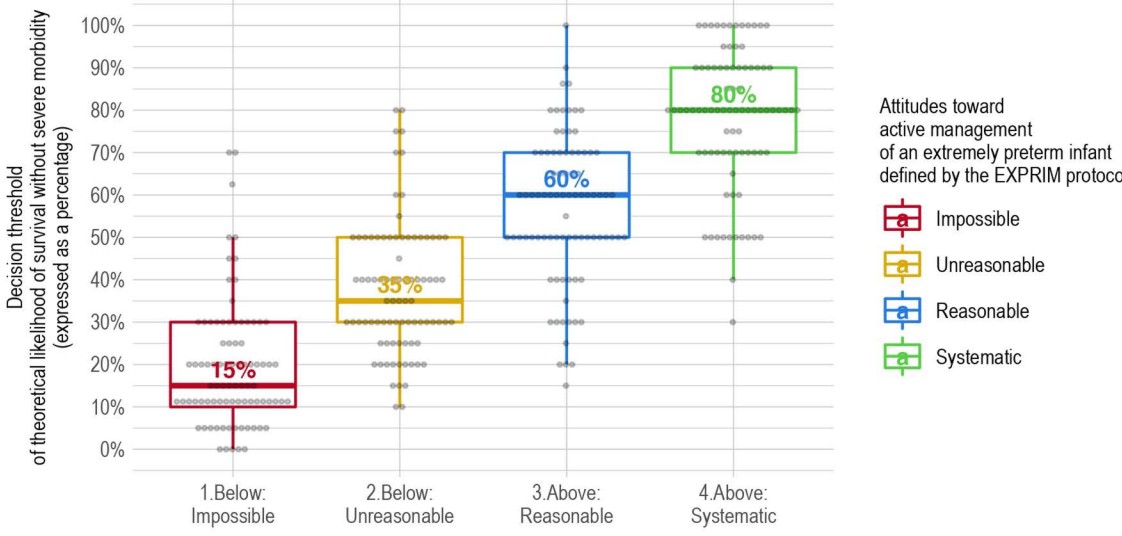

Boxplots – Median, 1st and 3rd quartiles, outlying values (in color) and individual opinion (in gray)

**Fig 2. Decision threshold for active management for each of the four attitudes for all respondents.**

thus felt that if the likelihood of survival without severe morbidity was less than 10%, active management was impossible, another quarter that it was impossible when the likelihood of survival was less than 30%. Five respondents considered that it was always possible and set

their threshold at 0%. The respondent most reluctant to propose active management considered it impossible unless the child had at least a 70% chance of survival without severe morbidity.

The median threshold of survival without severe morbidity below which active management was assessed as unreasonable (but possible if the parents wanted it) was 35% (IQR 30%-50%). Two respondents placed this threshold at 10% and two others at 80%.

The median threshold of survival without severe morbidity above which active management was considered reasonable (but so was palliative care if the parents preferred) was 60% (IQR 50%-70%). Because one respondent placed this threshold at 15% and another at 100%.

Finally, the median threshold of survival without severe morbidity above which respondents thought management should be considered systematic was 80% (IQR 70%-90%). One respondent placed this threshold at 30% and 12 others at 100%.

## Associations between individual determinants and decision thresholds

Median thresholds for the reasonable and systematic attitudes were significantly lower for the pediatricians than for the obstetricians and midwives, respectively 45% vs 50% and 60%, $P = .005$, and 70% vs 80% and 80%, $P = .005$. Results for the impossible and unreasonable attitudes did not differ significantly according to the professional category. The overall trend showed that thresholds of physicians, either pediatricians or obstetricians, were always either equal to or lower than those of midwives, regardless of the attitude considered (Fig 3).

In comparing thresholds according to professionals' sex (after excluding midwives), we found significant differences between decision thresholds for woman and man respondents for the attitudes of impossible, unreasonable, and reasonable, respectively 20% vs 7.5%, $P = .04$; 40% vs 30%, $P = .05$; and 40% vs 60%, $P = .02$. In our study, women chose not to propose active

**Fig 3.  Decision threshold for active management for each of the four attitudes by professional category.**

management — that is, considered it an impossible or unreasonable attitude — at higher thresholds than those chosen by men. Similarly, they chose active management — as a reasonable and systematic attitude — at higher thresholds than men (Fig 4).

It remained the same after midwives were excluded, but differences were no longer significant for any of these attitudes (S1 Fig).

Comparing decision thresholds of professionals with and without children showed no significant differences (S2 Fig). Nor did we find any difference according to age (S3 Fig) or professional experience (S4 Fig) for any of the four attitudes.

## Association between estimated outcomes of EPI and decision thresholds

Table 2 presents the distribution of professionals' estimated outcomes for EPI as percentages of survival without severe neonatal morbidity, for the five clinical situations considered (differing by gestational age and birthweight). For a fetus born at $23^{0/7}$ –$23^{3/7}$ weeks, weighing 500–600 g, and receiving optimal care, the median estimated survival without severe morbidity was 10%. A quarter of the respondents estimated this chance of survival without severe morbidity at 0% to 1%, and another quarter at 20% to 62.5%.

Fig 5 presents the association between professionals' estimated outcomes for EPI and their decision thresholds for active management for the first clinical situation. It shows the median decision thresholds for each professionals' quartile of estimated outcome for EPI born from $23^{0/7}$ through $23^{3/7}$ weeks and weighing from 500 to 600 g for each of the four attitudes. These thresholds did not differ statistically between the quartiles: the median threshold of survival without severe morbidity above which active management was considered reasonable was the same (60%) for the professionals in the first three quartiles of estimated outcome and even

Fig 4. Decision threshold for active management for each of the four attitudes by respondent's sex.

**Table 2. Distribution of the professionals' estimates of the outcomes of extremely preterm infants. Expressed as percentages of survival without severe neonatal morbidity.**

| Clinical situations | Median (%) | IQR (%) | Min; Max (%) |
|---|---|---|---|
| 23[0/7]–23[3/7] weeks, 500–600 g[1] | 10.0 | 1.0; 20.0 | 0.0; 62.5 |
| 23[4/7]–23[6/7] weeks, 500–600 g | 15.0 | 5.0; 25.0 | 0.0; 62.5 |
| 24[0/7]–24[3/7] weeks, 600–700 g[1] | 30.0 | 15.0; 45.0 | 0.6; 80.0 |
| 24[4/7]–24[6/7] weeks, 600–700 g[1] | 40.0 | 26.9; 55.8 | 2.5; 80.0 |
| 25[0/6]–25[6/7] weeks, > 700 g[1] | 60.0 | 45.0; 75.0 | 6.3; 100.0 |

*170 responses per situation.*

*[1]Two non-responses*

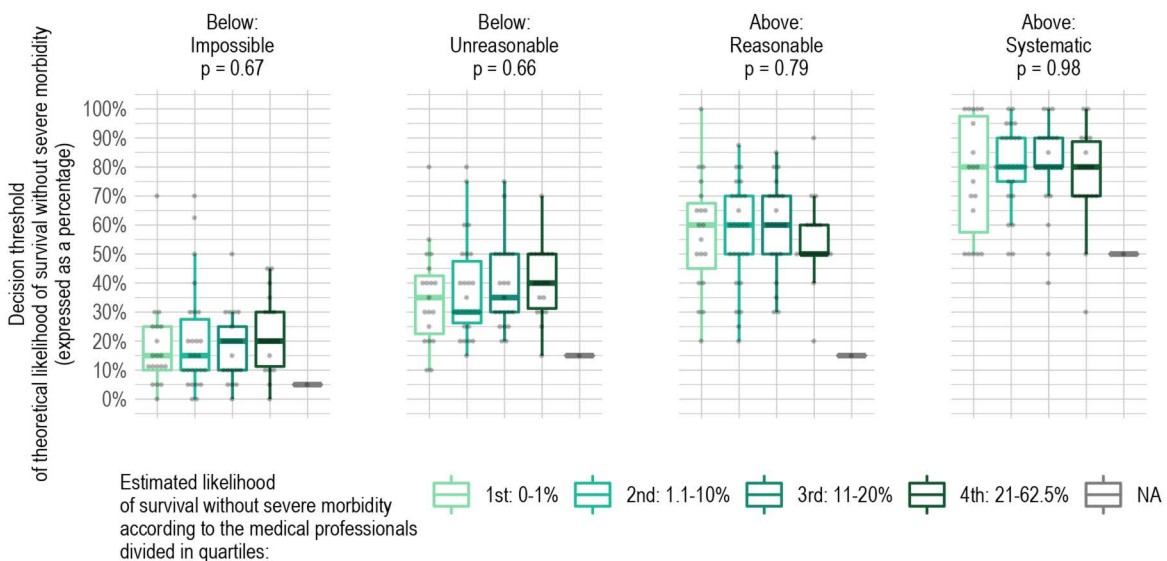

**Fig 5. Decision threshold for active management according to the professionals' estimates of survival without severe morbidity. In this figure, the professionals' estimates of the likelihood of survival without severe morbidity were expressed for an infant born from 23(0/7)–23(3/7) weeks weighing 500–600 grams.**

lower (50%) for those anticipating the best outcomes (fourth quartile). These thresholds did not differ either for the other clinical situations (S5-S8 Figs).

## Discussion

The objective of our study was to describe the interpersonal variability among professionals when deciding on active management for extremely preterm infants and to identify the factors contributing to this variability. Our findings revealed substantial variability in decision-making thresholds, expressed in terms of survival without severe neonatal morbidity, within the same medical team. Through participant interviews, we indeed identified thresholds with wide interquartile ranges (IQRs). For instance, one-quarter of respondents set their threshold for considering active management impossible at a survival likelihood of less than 30%

without severe morbidity, whereas another quarter only reached this decision when survival chances dropped below 10% without severe morbidity. This variability appeared to be influenced by profession and gender but were not associated with factors such as having children, age, experience, or the personal estimates of the neonates' outcomes. We also noted significant variability in the personal estimates of survival without morbidity for extremely preterm infants.

Wilkinson et al.[24] used prognostic thresholds to compare decisions about active management in three European countries — Sweden, England, and the Netherlands. In their study, the median thresholds of survival without severe sequelae below which active management seemed impossible were respectively around 10%, 5%, and 5%, while those above which active management should be systematic were 50%, 30%, and 25%. These thresholds were clearly higher among the French health care professionals in our unit. The median threshold below which active management would be considered impossible was 15% (IQR 10%-30%) while that above which it would be systematic was 80% (IQR 70%-90%). These results are consistent with the lower rate of active management in France than in other countries of a similar socioeconomic level and with medical practices in Sweden where active management has long been provided on a much larger scale. Moreover, the British, in their last published guidelines,[20] also reported, for the first time in a publication of this type, indicative prognostic thresholds expressed as "risk of death or severe sequelae." The thresholds recommended are also lower than those found in our study; the threshold at which the British considered active management impossible was below 10% and that for systematic active management, above 50%. It is difficult to understand the reasons for these differences between countries. They may involve cultural differences related to prognostic uncertainty and risk of disability, but also to differences in the perception of disability itself. The place and weight of parental choice in decision-making probably differ as well. Finally, health resource availability and cost of care could play a role.

In the 1990s, US studies showed that care providers underestimated the outcome of EPI compared with data in the literature and that this underestimation resulted in less frequent active management [17,19,22]. However, in our study, although very variable, professionals' survival estimates were higher than the most recent French data (EPIPAGE-2, 2011) and aligned with rates in comparable countries, indicating no underestimation of EPI outcomes [27]. Additionally, in our study, professionals' decision thresholds did not differ significantly according to their outcome estimates.

Despite this, our study demonstrated substantial differences in professionals' estimates of neonatal outcomes, with a gap of 20 percentage points or more between the first and third quartiles across all five clinical scenarios. This degree of variability aligns with findings from a recent survey evaluating the interpretation of the new British guidelines by UK neonatal professionals, where estimates of survival and severe disability ranged significantly (5%–90%) between the respondents in different clinical situation [28]. Similar variability has also been documented in studies conducted in Australia [29] and the United States [30]. Although it is rare in our clinical practice to provide specific survival estimates to guide parental decision-making, these prognostic estimates nonetheless shape how healthcare professionals present the situation. These discrepancies raise questions regarding the accuracy and consistency of the information conveyed to parents: not all parents may receive the same information.

Defining a 'correct' estimate of survival and severe disability is complex. In the US, the National Institute of Child Health and Human Development (NICHD) has developed an online calculator that allows clinicians to estimate neonatal outcomes based on five factors: gestational age, fetal weight, sex, plurality, and antenatal steroid administration [31]. However, these estimates, drawn from U.S. data, may not apply to other populations, including

France and the UK. Moreover, this tool has been described as performing moderately well and it was shown that the hospital of birth contributed substantially to outcome predictions [32]. This highlights the need to develop tools that could be locally adapted to estimate those outcomes [33].

This study was not intended to impose a standardized opinion that should be the same for everyone. Nonetheless, allowing all participants to share their individual opinions might contribute to more open team discussions and more consensual decisions. A variation in rates of active prenatal management from 20% through 80% between French regions calls into question the principle of equality of access to care [21]. We might even envision using, as the British have recently done, indicative prognostic thresholds, collectively defined within networks or even at a national level, and disseminating them as guidelines [27]. However, if one wishes to rely on our data to help define these thresholds, it is important to note that this study focused on professionals deeply involved in the care of extremely preterm infants, who tend to assess the acceptability of sequelae and disabilities more critically than either parents or adolescents born extremely preterm [34]. Future studies and discussions around guidelines could benefit from involving both of these groups in defining thresholds, allowing for a comparison of their perspectives with those of professionals.

Participation rate was high (75%); in particular, all eligible obstetricians responded. In view of the multidisciplinary nature of perinatal management, we considered it important to question pediatricians, obstetricians, and midwives. Other studies have surveyed either obstetricians or pediatricians, but not both; and midwives' opinions have not been sought [4,6,24]. We also studied other individual determinants potentially involved in the variability of these decisions. Finally, for this study, we chose to ask respondents to think about their decisions by using prognostic thresholds that consider the child's likely outcome. This is an original approach based not on a single prognostic factor (gestational age), but instead and more rationally on an assessment of the child's health status, defined by a recognized indicator [33]. The quantitative nature of the indicator also makes comparisons possible, in this study but also during team discussions; it avoids more subjective terms such as "poor" or "good" outcome.

Limitations include the fact that the data were collected four years ago. However, since the then, the senior obstetricians and pediatricians responsible for decision-making in our department have not changed. The protocol in use (EXPRIM) has remained the same as the one described in our study and has not been modified. We therefore believe that decisions regarding the active management of EPI have likely not changed during this period and the time elapsed since data collection does not diminish the relevance of our findings.

The small size of this study limits the precision of the reported percentages, particularly for certain subgroups. We also lacked the statistical power to perform a multivariate analysis of the factors influencing these decisions. Additionally, as a single-center study, our findings are closely tied to the practices of our hospital and the national context.

Some professionals also found the concept of prognostic thresholds difficult to grasp, stating that they "don't think about it like that". The mode of decision is so commonly structured around gestational age that some respondents linked outcome only to that factor and have never considered doing otherwise. The use of this new tool thus requires a period of adaptation.

## Conclusions

Using an approach based on the estimated overall health status revealed significant variability in individual decision thresholds for active management of EPI among professionals. These thresholds appear to be notably higher in our study than in other comparable countries, which may explain why active management is proposed less frequently before 26 weeks in France. Recognizing this variability and being able to compare personal thresholds with peers could

foster more consensus-based decision-making within teams. This could also lead to the development of collectively defined prognostic thresholds as guidelines. Future research, including input from parents and adults born extremely preterm, would be valuable for refining these thresholds and broadening the discussion. Additionally, tools to help clinicians better grasp the complexity of outcome estimations for EPI at a local level are needed.

## Supporting information

**S1 Fig. Decision threshold by respondent's sex (excluding midwives).** For active management for each of the four attitudes in the EXPRIM protocol.
(TIF)

**S2 Fig. Decision threshold according to whether the respondent has a child.** For active management for each of the four attitudes in the EXPRIM protocol.
(TIF)

**S3 Fig. Decision threshold by respondent's age.** For active management for each of the four attitudes in the EXPRIM protocol.
(TIF)

**S4 Fig. Decision threshold by respondent's professional experience.** For active management for each of the four attitudes in the EXPRIM protocol.
(TIF)

**S5 Fig. Decision threshold according to the professionals' estimates of the likelihood of survival without severe morbidity for an infant born from 23(4/7)–23(6/7) weeks weighing 500–600 grams.** For active management for each of the four attitudes in the EXPRIM protocol.
(TIF)

**S6 Fig. Decision threshold according to the professionals' estimates of the likelihood of survival without severe morbidity for an infant born from 24(0/7)–24(3/7) weeks weighing 600–700 grams.** For active management for each of the four attitudes in the EXPRIM protocol.
(TIF)

**S7 Fig. Decision threshold according to the professionals' estimates of the likelihood of survival without severe morbidity for an infant born from 24(4/7)–23(6/7) weeks weighing 600–700 grams.** For active management for each of the four attitudes in the EXPRIM protocol.
(TIF)

**S8 Fig. Decision threshold according to the professionals' estimates of the likelihood of survival without severe morbidity for an infant born from 25(0/7)–25(6/7) weeks weighing more than 700 grams** . For active management for each of the four attitudes in the EXPRIM protocol.
(TIF)

**S9 Appendix. Text of the self-administered questionnaire.**
(DOCX)

## Acknowledgements

Thank you to all the perinatal professionals who took the time to respond to this study.

## Author contributions

**Conceptualization:** Charlotte Girard, François Goffinet.

**Data curation:** Charlotte Girard.

**Formal analysis:** Charlotte Girard.

**Investigation:** Charlotte Girard.

**Methodology:** Hélène Collinot, Pierre-Henri Jarreau, François Goffinet.

**Supervision:** Hélène Collinot, François Goffinet.

**Visualization:** Charlotte Girard.

**Writing – original draft:** Charlotte Girard.

**Writing – review & editing:** Charlotte Girard, Hélène Collinot, Héloïse Torchin, Clara Rollet, Pierre-Henri Jarreau, François Goffinet.

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
