## [Decision Letter · Decision Letter 0]

22 Sep 2024

PONE-D-24-19296Variability and determinants of medical decisions thresholds to provide active management for extremely preterm infantsPLOS ONE

Dear Dr. Girard,

Thank you for submitting your manuscript to PLOS ONE. After careful consideration, we feel that it has merit but does not fully meet PLOS ONE’s publication criteria as it currently stands. Therefore, we invite you to submit a revised version of the manuscript that addresses the points raised during the review process.

We look forward to receiving your revised manuscript.

Kind regards,

Awol Yemane Legesse

Academic Editor

PLOS ONE

2. In the ethics statement in the Methods, you have specified that verbal consent was obtained. Please provide additional details regarding how this consent was documented and witnessed, and state whether this was approved by the IRB.

Additional Editor Comments:

The research question is relevant and timely. The study addresses the ethical complexities involved in deciding whether to provide active management for extremely preterm neonates. The findings can inform national and international guidelines on the management of extremely preterm neonates. This can lead to standardized care plans reducing variabilities with possibility of extending survival and reducing long term morbidities. Furthermore, it is important for parental involvement and resource allocation. The criterion showed differences in professionals' individual decision can be an eye opener for further discussions and policy worldwide. Though the study is a single centre study; overall, this manuscript contributes to the field of neonatology by providing insights into the complex factors that influence medical decisions for extremely preterm neonates. The manuscript is well written, applaud the authors for this outcome.

Please revise the title to add a geographic locator, such as "Level 3 Hospital, France," as this is a single-centre study.

I have a concern about this study. The data is a bit old as it was collected 4 years ago. In the fast-paced field of medicine, data should ideally be published within 2-3 years. However, the theme is less explored, so we consider your paper worth pursuing. You need to mention this in the limitations section.

Please revise the term infant with neonate throughout the manuscript as it is more appropriate terminology worldwide.

You need to revise the outline of the discussion section. The first paragraph should focus on the objective of the study and restate the main findings. The second paragraph should compare the findings with prognostic thresholds reported in the literature. The third paragraph should discuss the association between estimated outcome and the decision to provide active management. The fourth paragraph should cover the implications of the findings. The fifth paragraph should address the strengths and limitations of the study. Finally, the last paragraph should present the conclusions. Please remove the subheadings except for the conclusions.

Please clearly indicate your recommendations for clinical practice and future studies in your discussion section, along with the conclusions.

Consider the following article.

Wood K, Di Stefano LM, Mactier H, et al Individualised decision making: interpretation of risk for extremely preterm infants—a survey of UK neonatal professionals Archives of Disease in Childhood - Fetal and Neonatal Edition 2022;107:281-288. https://fn.bmj.com/content/107/3/281.info

Thank you

Reviewers' comments:

Reviewer's Responses to Questions

**Comments to the Author**

1. Is the manuscript technically sound, and do the data support the conclusions?

Reviewer #1: Yes

2. Has the statistical analysis been performed appropriately and rigorously? 

Reviewer #1: Yes

3. Have the authors made all data underlying the findings in their manuscript fully available?

Reviewer #1: Yes

4. Is the manuscript presented in an intelligible fashion and written in standard English?

Reviewer #1: Yes

5. Review Comments to the Author

Reviewer #1: Esteemed;

Thank you. I found your paper work good learning opportunity on a sensitive and practical issue having significant differences in consideration despite similarity of developmental level amongst nations, including those in Europe.

Based on the criteria requisites for publication on PLOS ONE (1-7), I found it suitable for publication with minor revision, or explanations:

1. Data collection had finalized in April 2020, and the time lapsed towards publication concerns me; four years lapsed. I am worried this NICU level 3 might have changed its protocol/ approach by now.

2. Methods and materials: I prefer methods only as a topic than methods and materials. There was not any material experimentation.

3. Table 1 contains limited data (can leave this comment if only mine amongst reviewers).

4. Results: table 1 years of work experience - we use 3 years as least experienced; is two years a cut point per French MOH? Some professions use five (5) years.

6. PLOS authors have the option to publish the peer review history of their article (what does this mean? ). If published, this will include your full peer review and any attached files.

**Do you want your identity to be public for this peer review?** For information about this choice, including consent withdrawal, please see our Privacy Policy .

Reviewer #1: No

---

## [Author Response · Author response to Decision Letter 0]

15 Oct 2024

We would like to thank Dr Awol Yemane Legesse, Academic Editor at Plos One, and Reviewer#1 for their constructive comments and suggestions, which will enhance the quality of our manuscript. Below is our point-by-point response. The pages and lines refer to the final manuscript.

1. Journal Requirements

Comments to the Author: Please ensure that your manuscript meets PLOS ONE's style requirements, including those for file naming.

Author’s answer : We have made the relevant checks

Comments to the Author: In the ethics statement in the Methods, you have specified that verbal consent was obtained. Please provide additional details regarding how this consent was documented and witnessed, and state whether this was approved by the IRB.

Author’s answer : In the ethics statement in the Methods, we replaced the phrase “Each person interviewed gave oral consent” with this paragraph “Consent was obtained orally from each participant by the principal investigator prior to answering the questionnaire. An information and non-opposition notice for the use of data was written with the project and was made available to the participants. This notice was provided to the research ethics committee that studied our protocol and contributed to its evaluation.” (page 3, line 62 to 65) The information and non-opposition notice was added to the submission files.

2. Additional Editor Comments

Comments to the Author: The research question is relevant and timely. The study addresses the ethical complexities involved in deciding whether to provide active management for extremely preterm neonates. The findings can inform national and international guidelines on the management of extremely preterm neonates. This can lead to standardized care plans reducing variabilities with possibility of extending survival and reducing long term morbidities. Furthermore, it is important for parental involvement and resource allocation. The criterion showed differences in professionals' individual decision can be an eye opener for further discussions and policy worldwide. Though the study is a single center study; overall, this manuscript contributes to the field of neonatology by providing insights into the complex factors that influence medical decisions for extremely preterm neonates. The manuscript is well written, applaud the authors for this outcome.

Author’s answer : We thank the editor for this encouraging comment, which supports us in our effort to publish this work. We indeed believe that, although it is a single-center study, it opens up a discussion that is much more universal.

Comments to the Author: Please revise the title to add a geographic locator, such as "Level 3 Hospital, France," as this is a single-centre study.

Author’s answer : We propose the following new title: "Variability and Determinants of Medical Decision Thresholds for Active Management of Extremely Preterm Infants in a Level 3 Hospital in France."

Comments to the Author: I have a concern about this study. The data is a bit old as it was collected 4 years ago. In the fast-paced field of medicine, data should ideally be published within 2-3 years. However, the theme is less explored, so we consider your paper worth pursuing. You need to mention this in the limitations section.

Author’s answer : Thank you for raising this important concern. We acknowledge that the data is somewhat outdated, and this is indeed one of the limitations of our study. We have incorporated this point into the limitations section (page 19-20, line 368-373).

“Limitations include the fact that the data were collected four years ago. However, since the then, the senior obstetricians and pediatricians responsible for decision-making in our department have not changed. The protocol in use (EXPRIM) has remained the same as the one described in our study and has not been modified. We therefore believe that decisions regarding the active management of EPI have likely not changed during this period and the time elapsed since data collection does not diminish the relevance of our findings”.

Comments to the Author: Please revise the term infant with neonate throughout the manuscript as it is more appropriate terminology worldwide.

Author’s answer : We appreciate the editor's clarification and his efforts to promote the international comprehension of our article. However, we would like to raise a question regarding this suggestion. The MeSH thesaurus selected the term “Extremely Preterm Infant” as a keyword and the majority of articles on this topic refer to these neonates using this same terminology. For instance, Bell EF et al. used the term “Extremely Preterm Infant” in the title of their study “Mortality, In-Hospital Morbidity, Care Practices, and 2-Year Outcomes for Extremely Preterm Infants in the US, 2013-2018,” published in JAMA in 2022. With your approval, we would prefer to continue using this term but we can certainly replace it if you prefer.

Comments to the Author: You need to revise the outline of the discussion section :

- The first paragraph should focus on the objective of the study and restate the main findings.

- The second paragraph should compare the findings with prognostic thresholds reported in the literature.

- The third paragraph should discuss the association between estimated outcome and the decision to provide active management.

- The fourth paragraph should cover the implications of the findings.

- The fifth paragraph should address the strengths and limitations of the study.

- Finally, the last paragraph should present the conclusions. Please clearly indicate your recommendations for clinical practice and future studies in your discussion section, along with the conclusions.

Please remove the subheadings except for the conclusions.

Author’s answer : We appreciate the editor's valuable advice, which has helped us enhance the clarity and impact of our discussion. We have removed the subheadings and reorganized the paragraphs to better present the information (page 16 to 20).

In particular, we :

- Focused on the objective of the study and restated the main findings in the first paragraph (page 16, line 291 to 298): “The objective of our study was to describe the variability in individual thresholds among professionals when deciding on active management for extremely preterm infants (before 26 weeks of gestation) and to identify the factors contributing to this variability, particularly professionals' estimates of these neonates' outcomes. Our findings revealed substantial variability in decision-making thresholds among professionals within the same medical team. These differences appeared to be influenced by profession and gender but were not associated with factors such as having children, age, experience, or their estimates of these neonates' outcomes. We also noted significant variability in the estimates of survival without morbidity for extremely preterm infants.”

- Clearly articulated our recommendations for clinical practice and future studies in our discussion section (page 19, line 348 to 355): “We might even envision using, as the British have recently done, indicative prognostic thresholds, collectively defined within networks or even at a national level, and disseminating them as guidelines. However, if one wishes to rely on our data to help define these thresholds, it is important to note that this study focused on professionals deeply involved in the care of extremely preterm infants, who tend to assess the acceptability of sequelae and disabilities more critically than either parents or adolescents born extremely preterm.[34] Future studies and discussions around guidelines could benefit from involving both of these groups in defining thresholds, allowing for a comparison of their perspectives with those of professionals.”

- Clearly articulated our recommendations for clinical practice and future studies in the conclusions (page 20, line 389 to 393): “This could also lead to the development of collectively defined prognostic thresholds as guidelines. Future research, including input from parents and adults born extremely preterm, would be valuable for refining these thresholds and broadening the discussion. Additionally, tools to help clinicians better grasp the complexity of outcome estimations for EPI at a local level are needed.”

Comments to the Author: Consider the following article.

Wood K, Di Stefano LM, Mactier H, et al Individualized decision making: interpretation of risk for extremely preterm infants—a survey of UK neonatal professionals Archives of Disease in Childhood - Fetal and Neonatal Edition 2022;107:281-288. https://fn.bmj.com/content/107/3/281.info

Author’s answer : We thank the editor for recommending this article, which we have read carefully. Its conclusions are indeed very insightful and complement our study well. We have cited this work in our discussion (page 18, line 325 to 330).

“Despite this, our study demonstrated substantial differences in professionals' estimates of neonatal outcomes, with a gap of 20 percentage points or more between the first and third quartiles across all five clinical scenarios. This degree of variability aligns with findings from a recent survey evaluating the interpretation of the new British guidelines by UK neonatal professionals, where estimates of survival and severe disability ranged significantly (5%–90%) between the respondents in different clinical situation.”

We also added a new point addressing the variability in the estimates of neonatal outcomes that were discussed in this article and that we were also able to measure (page 18, line 330-342).

“Similar variability has also been documented in studies conducted in Australia and the United States. Although it is rare in our clinical practice to provide specific survival estimates to guide parental decision-making, these prognostic estimates nonetheless shape how healthcare professionals present the situation. These discrepancies raise questions regarding the accuracy and consistency of the information conveyed to parents: not all parents may receive the same information.

Defining a ‘correct’ estimate of survival and severe disability is complex. In the US, the National Institute of Child Health and Human Development (NICHD) has developed an online calculator that allows clinicians to estimate neonatal outcomes based on five factors: gestational age, fetal weight, sex, plurality, and antenatal steroid administration. However, these estimates, drawn from U.S. data, may not apply to other populations, including France and the UK. Moreover, this tool has been described as performing moderately well and it was shown that the hospital of birth contributed substantially to outcome predictions. This highlights the need to develop tools that could be locally adapted to estimate those outcomes.”

3. Reviewer#1’s comments

Comments to the Author: I found your paper work good learning opportunity on a sensitive and practical issue having significant differences in consideration despite similarity of developmental level amongst nations, including those in Europe.

Author’s answer : We express our gratitude to Reviewer #1 for their interest in our article and for their thoughtful consideration of this topic, which is of great significance to us.

Comments to the Author: Data collection had finalized in April 2020, and the time lapsed towards publication concerns me; four years lapsed. I am worried this NICU level 3 might have changed its protocol/ approach by now.

Author’s answer : Thank you for raising this important concern, which has also been addressed by the editor. We acknowledge that this time lapse is indeed one of the limitations of our study and we have incorporated this point into the limitations section (page 19-20, line 368-373).

“Limitations include the fact that the data were collected four years ago. However, since the then, the senior obstetricians and pediatricians responsible for decision-making in our department have not changed. The protocol in use (EXPRIM) has remained the same as the one described in our study and has not been modified. We therefore believe that decisions regarding the active management of EPI have likely not changed during this period and the time elapsed since data collection does not diminish the relevance of our findings”.

Comments to the Author: Methods and materials: I prefer methods only as a topic than methods and materials. There was not any material experimentation.

Author’s answer : Thank you for this relevant comment. We have modified the section title accordingly.

Comments to the Author: Table 1 contains limited data (can leave this comment if only mine amongst reviewers).

Author’s answer : We understand your comment. We indeed collected little personal information about the respondents. There are several reasons for this: first, the small sample size required that our categories be broad enough to have sufficient numbers for usable statistics. Additionally, unfortunately, research regulations in France do not allow us to request certain personal data that initially seemed very interesting, such as religious orientation or ethnic backgrounds. We hope this response addresses your concern.

Comments to the Author: Results: table 1 years of work experience - we use 3 years as least experienced; is two years a cut point per French MOH? Some professions use five (5) years.

Author’s answer : Thank you for your question. We hope our response meets your expectations: We did not select this cutoff point based solely on a specific French practice but rather by considering the size of our categories and their practical relevance. We believe that the distinction between two and ten years of work experience is significant in terms of clinical experience and could yield valuable insights. As mentioned earlier, our small sample size necessitated that our categories be broad enough to ensure sufficient numbers for meaningful statistical analysis.

---

## [Decision Letter · Decision Letter 1]

5 Jan 2025

PONE-D-24-19296R1Variability and determinants of medical decisions thresholds for active management of extremely preterm infants in a level 3 hospital in FrancePLOS ONE

Dear Dr.  Girard,

Thank you for submitting your manuscript to PLOS ONE. After careful consideration, we feel that it has merit but does not fully meet PLOS ONE’s publication criteria as it currently stands. Therefore, we invite you to submit a revised version of the manuscript that addresses the points raised during the review process.

We look forward to receiving your revised manuscript.

Kind regards,

Awol Yemane Legesse

Academic Editor

PLOS ONE

Additional Editor Comments:

Dear Authors, though you have improved the manuscript, please address the comments and suggestions of the second reviewer. Additionally, in the introduction section, please mention the rationale for why you are conducting the study by stating the specific threshold and specific survival rate reported elsewhere that you intend to compare.

Reviewers' comments:

Reviewer's Responses to Questions

**Comments to the Author**

1. If the authors have adequately addressed your comments raised in a previous round of review and you feel that this manuscript is now acceptable for publication, you may indicate that here to bypass the “Comments to the Author” section, enter your conflict of interest statement in the “Confidential to Editor” section, and submit your "Accept" recommendation.

Reviewer #2: (No Response)

2. Is the manuscript technically sound, and do the data support the conclusions?

Reviewer #2: Yes

3. Has the statistical analysis been performed appropriately and rigorously? 

Reviewer #2: Yes

4. Have the authors made all data underlying the findings in their manuscript fully available?

Reviewer #2: Yes

5. Is the manuscript presented in an intelligible fashion and written in standard English?

Reviewer #2: No

6. Review Comments to the Author

Reviewer #2: (No Response)

7. PLOS authors have the option to publish the peer review history of their article (what does this mean? ). If published, this will include your full peer review and any attached files.

**Do you want your identity to be public for this peer review?** For information about this choice, including consent withdrawal, please see our Privacy Policy .

Reviewer #2: **Yes: ** Hale Teka

---

## [Author Response · Author response to Decision Letter 1]

11 Feb 2025

We would like to thank Dr. Awol Yemane Legesse, Academic Editor at PLOS ONE, and Dr. Hale Teka for their constructive comments and suggestions on the second version of our PLOS ONE manuscript. We have carefully considered their feedback, and we hope that our revisions will improve both the quality and readability of the article.

Below, we provide our point-by-point responses.

The page and line numbers refer to the final version of the manuscript (without the track changes).

1. Additional Editor Comments

Comments to the Author: In the introduction section, please mention the rationale for why you are conducting the study by stating the specific threshold and specific survival rate reported elsewhere that you intend to compare.

Author’s answer : We appreciate your thoughtful suggestion.

However, we would like to clarify that the main aim of our study was not to compare our thresholds with those reported in the literature. But rather to use this particular framework within our specific context to examine the variability of medical decisions.

We clarified this by adding phrases in the introduction section:

- Line 87-89: Their study was the first in which neonatal physicians’ views about prognosis-based thresholds for resuscitation were assessed and they did not issue any recommendations [24]. Such thresholds are not routinely used and have never been assessed in France.

- Line 92-95: To achieve this, we adopted an approach similar to that of Wilkinson et al., basing decision-making on prognostic thresholds expressed as survival without severe disability. We applied this framework to the management decisions for EPI (born before 26 WG) in a single obstetrics-pediatric unit.

The comparison with reported thresholds was incorporated in the discussion section, with the goal of contextualizing our results internationally and comparing our findings with existing data. We are concerned that introducing additional numerical details in the introduction may unnecessarily complicate an already complex topic.

2. Reviewers’ comments

Comments to the Author: Style. I understand that this article might have been submitted to another journal. Please revise the format according to the PLoS one authors guideline. There are some sections which does not conform to the style of PLoS one's style. For example, key words should come immediately after conclusion, no need of article summary, what is known about this subject, what this study adds etc. Please revisit your style of manuscript writing.

Author’s answer : Thank you very much for your careful review of the style.

This was done:

- The sections “Ethics approval and consent to participate” and “ Availability of data and materials” were relocated and directly integrated into the Methods section (Line 179-191).

- The “Keywords” section was moved to follow the Conclusions (Line 367-369).

- The superfluous sections (“Author’s contribution”, “Article Summary”, “What’s Known on This Subject”, “What This Study Adds”, “List of abbreviations”) were entirely removed.

Comments to the Author: Title. The title is confusing as it stands now. Please improve it. Propose titles: "Healthcare Professionals Interpersonal Variability in Medical Decision Thresholds for Active Management of Extremely Preterm Neonates in X Hospital, a level 3 healthcare institution in France."

Author’s answer : Thank you for your feedback and suggestion.

We propose the following title: “Healthcare Professionals Interpersonal variability and determinants of medical decision thresholds for active management of extremely preterm infants in a level 3 perinatal center in France “

Comments to the Author: Abstract. The results section of the abstract is confusing. If readers decide to read just your abstract "The median threshold (threshold of what?) below which active management was considered not acceptable was 15% (IQR 10-30%). The median threshold (threshold of what?) above which active management could not be refused was 80% (IQR 70- 90%). It varied significantly by profession (p=0.005)." does not give meaning. I understand what you meant after reading the whole manuscript but reading just the abstract, does not give any meaning. Please revise it.

Author’s answer : Thank you very much for this comment, which will help make our abstract clearer. We revise the Results section of the abstract accordingly (Line 38 to 43)

“The median threshold of survival without severe neonatal morbidity below which active management was deemed impossible was 15% (IQR 10-30%), while the median threshold above which active management could not be refused was 80% (IQR 70-90%)”

Comments to the Author: Abstract. In general, your abstract does not reflect the main findings of you manuscript.

Author’s answer : Thank you for raising this point.

We have revised our entire abstract and have particularly strengthened the Results and Conclusions section to ensure it aligns more effectively with the content of our manuscript (Line 24 to 49) :

- “Results

85 (75%) eligible professionals responded. The median threshold of survival without severe neonatal morbidity below which active management was deemed impossible was 15% (IQR 10-30%), while the median threshold above which active management could not be refused was 80% (IQR 70-90%). Wide IQRs indicated significant variability in individual thresholds. This variability appeared to be influenced by profession and gender but was not associated with factors such as having children, age, experience, or the personal estimates of the neonates' outcomes.”

- “ Conclusions

Decision thresholds for active management of EPI, expressed in terms of survival without severe neonatal morbidity, vary significantly among professionals. The thresholds reported in our study were notably higher than those observed in other countries, which may help explain the lower rates of active management before 26 weeks in France. Recognizing these differences and comparing personal thresholds with peers could facilitate more consensus-based decision-making within teams.”

Comments to the Author: Keywords. Your title must be formed by your key words. In your article, the keywords are not reflected in your title. Please revise your keywords accordingly.

Author’s answer :

We initially selected our keywords from the MeSH thesaurus, although it doesn't offer a wide range of options. In addition to the thesaurus, we selected the following keywords (Line 368-369):

- “Extremely Preterm Infant”

- “Medical Decisions”

- “Survival Thresholds”

- “Interpersonal Variability”

- “Palliative Care vs Active Treatment”

Comments to the Author: Methods. In your methods section, the way you classifed age groups and levels of experience sounds random. What is the basis for classifying age groups and level of exprience for your study? Please define the years for junior, mid level, and senior based on evidence and then classify.

Author’s answer :

In our study, the categorization was not based on the literature. For both the age groups and levels of experience, we structured the three categories to ensure homogeneous group sizes, with the aim of increasing statistical power.

This clarification has been added in the Methods (Line 111-112):

“The categories were defined to ensure homogeneous group sizes, with the aim of increasing statistical power.”

Comments to the Author: Results. In the methods section you stated that experience was categorized as <2, 2-4, and >4 years but in Table 1 it is different. Pls be consistent.

Author’s answer : Thank you for catching on this regretful mistake.

This was corrected in the Methods section (numbers in Table 1 are correct) (Line 110-111):

“We analyzed the latter two characteristics in three categories: age as less than 30 years, 30–40 years, and older than 40 years, and professional as less than 2 years, 2–10 years, and longer than 10 years.”

Comments to the Author: Results. Line 234" Five respondents considered that it was never impossible and set their threshold at 0%." Please rewrite this. I understand it but instead of using double negation, use a simple statement.

Author’s answer :

This was done. We changed the phrasing (Line 206).

"Five respondents considered that it was always possible and set their threshold at 0%."

Comments to the Author: Discussion. Please use the first paragraph to restate the aim, and summarize your findings including the numerical values of the relevant findings that need to be discussed.

Author’s answer :

We have modified the first paragraph of the discussion section and specifically added numerical values. We hope that this paragraph is now more impactful. Line 261-271:

“The objective of our study was to describe the interpersonal variability among professionals when deciding on active management for extremely preterm infants and to identify the factors contributing to this variability. Our findings revealed substantial variability in decision-making thresholds, expressed in terms of survival without severe neonatal morbidity, within the same medical team. Through participant interviews, we indeed identified thresholds with wide interquartile ranges (IQRs). For instance, one-quarter of respondents set their threshold for considering active management impossible at a survival likelihood of less than 30% without severe morbidity, whereas another quarter only reached this decision when survival chances dropped below 10% without severe morbidity. This variability appeared to be influenced by profession and gender but were not associated with factors such as having children, age, experience, or the personal estimates of the neonates' outcomes. We also noted significant variability in the personal estimates of survival without morbidity for extremely preterm infants.”

Comments to the Author: Discussion. In subsequent paragraphs, please discuss each finding you summarized in the first paragraph to compare and contrast, state the reason for similarities or differences. The paragraph’s are lengthy and lacks clarity. Your discussion section needs a major revision.

Author’s answer : Thank you for your comment. Regarding the structure of our discussion, we would like to highlight that during the previous revision, thanks to the editor’s numerous comments, we already conducted an in-depth revision of this section.

We specifically followed the outline provided by the editor at that time.

“You need to revise the outline of the discussion section :

- The first paragraph should focus on the objective of the study and restate the main findings.

- The second paragraph should compare the findings with prognostic thresholds reported in the literature.

- The third paragraph should discuss the association between estimated outcome and the decision to provide active management.

- The fourth paragraph should cover the implications of the findings.

- The fifth paragraph should address the strengths and limitations of the study.

- Finally, the last paragraph should present the conclusions. Please clearly indicate your recommendations for clinical practice and future studies in your discussion section, along with the conclusions.

- Please remove the subheadings except for the conclusions.”

We fully understand the importance of clarity and conciseness in discussing the results. However, we believe that the current structure meets the editor's expectations by addressing each key point in a logical and organized manner. If you have specific suggestions to improve clarity or reduce the length of the paragraphs, we would be happy to consider them for a further revision. With your consent, we would like to propose keeping the revised structure for now, but are open to further adjustments if necessary.

---

## [Editor Report · Decision Letter 2]

27 Feb 2025

Healthcare Professionals Interpersonal variability and determinants of medical decision thresholds for active management of extremely preterm infants in a level 3 perinatal center in France

PONE-D-24-19296R2

Dear Dr. Charlotte Girard,

We’re pleased to inform you that your manuscript has been judged scientifically suitable for publication and will be formally accepted for publication once it meets all outstanding technical requirements.

Kind regards,

Awol Yemane Legesse

Academic Editor

PLOS ONE
---

## [Editor Report · Acceptance letter]

PONE-D-24-19296R2

PLOS ONE

Dear Dr. Girard,

I'm pleased to inform you that your manuscript has been deemed suitable for publication in PLOS ONE. Congratulations! Your manuscript is now being handed over to our production team.

Kind regards,

on behalf of

Dr. Awol Yemane Legesse

Academic Editor

PLOS ONE